# OpenReview forum: "Research on Security Assessment and Safety Hazards Optimization of Large Language Models"
_IEEE.org/ICIST/2024/Conference — IEEE ICIST 2024 Conference Submission_

### Official Review · Reviewer_y9pR · 2024-08-21
**Review Results**

**Rating:** 5
**Confidence:** 4

**Review:**

This research assessed Chinese language model security, creating evaluation tools and comparing model performances. It identified strengths and risks across models and proposed the ST-GPT for enhanced security. The study advises legal compliance and continual AI improvement for all models.

This study identified potential safety risks and provided ST-GPT for each safety risk. Can we directly reduce the occurrence of such potential safety risks internally?

It is necessary for domestic and foreign models to comply with the legal framework of their respective countries, reduce AI hallucinations, continuously expand the corpus, and perform corresponding updates and iterations. So, how to achieve this?

---

### Official Review · Reviewer_WMVt · 2024-08-22
**The paper is logically clear, the simulation results are abundant.**

**Rating:** 6
**Confidence:** 3

**Review:**

This study investigates the performance of mainstream large language models in Chinese security generation tasks, explores the possible security risks of large language models, and proposes improvement strategies. There are the following questions need to be considered:
1. This study compares the performance of three models in 6 security tasks,  please provide a detailed introduction to the performance advantages of each of these models.
2. This study provides excellent data comparison results. However, the size of the graphics should be adjusted to ensure that readers can clearly see the contribution made.
3. Some grammar issues should be checked and corrected, such as "Llama 3" on page 7.

---

### Official Review · Reviewer_CcZ6 · 2024-08-27
**My Comments**

**Rating:** 7
**Confidence:** 3

**Review:**

The paper provides an examination of the security aspects of mainstream large language models like GPT-4, ERNIE Bot, and Claude, focusing on their application in Chinese security generation tasks. Here’s a detailed review and some specific suggestions:

1.	The author should provide a more comprehensive background of the problem space, referencing current trends and recent studies that highlight the need for this research.

2.	Ensure the literature review includes the most recent and relevant studies, particularly those published in the last 2-3 years, to demonstrate the timeliness of the research.

3.	It is suggested to discuss practical steps that can be taken by developers and policymakers based on your research findings to enhance model security.

---

### Decision · Program_Chairs · 2024-09-08

Accept (Oral)